

# Monitoring and benchmarking Earth system model simulations with ESMValTool v2.12.0

Axel Lauer[1], Lisa Bock[1], Birgit Hassler[1], Patrick Jöckel[1], Lukas Ruhe[2], Manuel Schlund[1]

[1]Deutsches Zentrum für Luft- und Raumfahrt (DLR), Institut für Physik der Atmosphäre, Oberpfaffenhofen, Germany
[2]University of Bremen, Institute of Environmental Physics (IUP), Bremen, Germany

*Correspondence to*: Axel Lauer (axel.lauer@dlr.de)

**Abstract.** Earth system models (ESMs) are important tools to improve our understanding of present-day climate and to project climate change under different plausible future scenarios. For this, ESMs are continuously improved and extended resulting in more complex models. Particularly during the model development phase, it is important to continuously monitor how well the historical climate is reproduced and to systematically analyze, evaluate, understand, and document possible shortcomings. For this, putting model biases relative to observations into the context of deviations shown by other state-of-the-art models greatly helps to assess which biases need to be addressed with higher priority. Here, we introduce the new capability of the open-source community-developed Earth System Model Evaluation Tool (ESMValTool) to monitor running or benchmark existing simulations with observations in the context of results from the Coupled Model Intercomparison Project (CMIP). To benchmark model output, ESMValTool calculates metrics such as the root-mean-square error, the Pearson correlation coefficient, or the Earth mover's distance relative to reference datasets. This is directly compared to the same metric calculated for an ensemble of models such as the one provided by CMIP6, which provides a statistical measure for the range of values that can be considered typical for state-of-the-art ESMs. Results are displayed in different types of plots such as map plots or time series with different techniques such as stippling (maps) or shading (time series) used to visualize the typical range of values for a given metric from the model ensemble used for comparison. Automatic downloading of CMIP results from the Earth System Grid Federation (ESGF) makes application of ESMValTool for benchmarking of individual model simulations, for example in preparation of CMIP7, easy and very user friendly.



## 1    Introduction

Earth System Models (ESMs) are complex numerical representations of the Earth system including not only interactions between the physical components such as atmospheric, oceanic, land and sea ice dynamics, but also climate relevant chemical and biological processes. In the past years, ESMs became essential tools to better understand the human impact on the climate system and to project future climate change under different emission scenarios.

For this, ESMs are continuously developed and improved with new processes added and existing processes described in more detail. As with any model development activity, a thorough evaluation of new model results is a fundamental prerequisite to assess the model's performance, and thus the model's suitability for a given scientific application (fitness for purpose). Evaluating ESMs has become quite complex as there is a growing multitude of relevant parameters from different Earth system components that typically require a team of scientists with different expertise to fully assess all details. Furthermore, evaluation of some parameters such as, for instance, biogeochemical components might suffer from a lack of global observations that are suitable for a comparison with the model results as these are very hard to obtain.

One possibility to quickly assess deviations from observations present in a new model simulation is to put them into perspective by comparing the biases with the ones obtained from a large number of other state-of-the-art ESMs. For this, for example results from the Coupled Model Intercomparison Phase 5 and 6 (Eyring et al., 2016; Taylor et al., 2012) can be used to get an overview of which biases can be considered "acceptable for now", and which would need more attention and more detailed analysis and comparisons with observations. The same approach can be used to monitor running model simulations to identify significant problems early on. In this article, we exemplary demonstrate the new capability of the Earth System Model Evaluation Tool (ESMValTool) to obtain a broad overview by benchmarking a given model simulation with CMIP results using different relevant diagnostics such as climatologies, seasonal and diurnal cycles, or geographical distributions. The examples are meant as a starting point and can be extended easily and applied to different components of the Earth system.



## 2 Methods

### 2.1 Earth System Model Evaluation Tool

The Earth System Model Evaluation Tool (ESMValTool) is an open-source community-developed diagnostics and performance metrics tool for the evaluation and analysis of Earth system models with Earth observations (Righi et al., 2020; Eyring et al., 2020; Lauer et al., 2020; Weigel et al., 2021). ESMValTool has been developed into a well-tested and well-documented tool that facilitates analysis across different Earth system components (e.g. atmosphere, ocean, land, sea ice).

While originally designed to facilitate a comprehensive and rapid evaluation of models participating in the Coupled Model Intercomparison Project, the tool can now also be used to analyze some output from regional models, a large variety of gridded observational data, and reanalysis datasets. Recent improvements include the possibility to read and process operational output of selected models produced by running a model through its standard workflow, without the requirement of applying further post-

processing steps as well as the strongly improved capability to handle unstructured grids (Schlund et al., 2023).

ESMValTool allows for consistent processing of all model and observational datasets such as, for instance, regridding to common grids, masking of land/sea and missing values, vertical interpolation, etc. This allows for a fair comparison of all diagnostics and metrics calculated for individual models with

each other. With the recently added features of being able to specify model datasets with wildcards and automatic download of datasets from the Earth System Grid Federation (ESGF), ESMValTool is well suited to provide the context for comparing model deviations from observations with each other in an easy and convenient way. This allows to check a large set of parameters and provides the flexibility to extend existing benchmarking "recipes" easily. Recipes are configuration files for ESMValTool that

define all input data, preprocessing steps, and diagnostics or metrics to be applied. All examples shown in this publication can be reproduced with ESMValTool version 2.12.0 using the recipes "recipe_lauer24gmd_fig*.yml" that are available on Zenodo (doi: 10.5281/zenodo.11198445).



## 2.2 Available metrics

For the purpose of assessing the general performance of a new model simulation and to quickly identify

potential problems that require more attention, a number of metrics such as, for instance, bias or root-mean-square error, are available that can be applied over one or multiple dimension coordinates of a dataset. These dimensions include longitude, latitude and time, and for parameters that are vertically resolved, also a vertical coordinate such as pressure or altitude. For example, consider two 3-dimensional datasets (model and reference) with dimensions time, latitude, and longitude. If a metric is applied over

the time dimension, the result is a 2-dimensional map with dimensions latitude and longitude; if a metric is applied over the horizontal dimensions latitude and longitude, the result is a 1-dimensional time series with dimension time.

For all metrics, an unweighted and weighted version exists. In the latter case, each point (in time and/or space) that enters the metric calculation is weighted with a factor $w_i$ (details on the calculations are given

in the corresponding sections below). In case a metric is calculated over time, the weights are defined as the length of the time intervals. If a metric is calculated over geographical coordinates (latitude and/or longitude), the grid box area size of each grid cell is used as weights. If a metric is calculated over time and geographical coordinates, the weights are calculated as the product of the above. Weights are normalized, i.e., $\sum_{i=1}^{N} w_i = 1$ ($N$: number of data points).

The following sections give an overview of the metrics that are available.

### 2.2.1 Absolute and relative BIAS

The absolute BIAS metric calculates the difference between a given dataset $X$ and a reference dataset $R$ (e.g. observations) as

$$BIAS_{abs} = X - R \tag{1}$$


The relative BIAS is obtained by dividing by the reference dataset $R$:



$$BIAS_{rel} = \frac{X - R}{R} \qquad (2)$$

In order to avoid spurious values as a result of very small values of $R$, an optional threshold to mask
values close to zero in the denominator can be provided.

### 2.2.2   Weighted and unweighted RMSE

The average root-mean-square error between a dataset $X$ and a reference dataset $R$ is calculated as

$$RMSE_{unweighted} = \sqrt{\frac{1}{N}\sum_{i=1}^{N}(X_i - R_i)^2} \qquad (3)$$

Here, $N$ gives the number of coordinate values over all dimensions over which the metric is applied.
Optionally, the individual values can be weighted with normalized weights $w_i$:

$$RMSE_{weighted} = \sqrt{\sum_{i=1}^{N} w_i(X_i - R_i)^2} \qquad (4)$$

A smaller RMSE corresponds to a better performance. More information on the weights is given at the
beginning of Section 2.2.

### 2.2.3   Weighted and unweighted Pearson correlation coefficient

The Pearson correlation coefficient ($r$) measures the linear correlation between two datasets and is defined
as the ratio between the covariance of two variables and the product of their standard deviations:





$$r_{unweighted} = \frac{\sum_{i=1}^{N}(X_i - \bar{X})(R_i - \bar{R})}{\sqrt{\sum_{i=1}^{N}(X_i - \bar{X})^2}\sqrt{\sum_{i=1}^{N}(R_i - \bar{R})^2}} \tag{5}$$


Here, $\bar{X}$ and $\bar{R}$ denote the average of the dataset X and R, respectively, over the selected dimension coordinate. Similar to the RMSE, the weighted $r$ considers normalized weights $w_i$:

$$r_{weighted} = \frac{\sum_{i=1}^{N}[w_i(X_i - \bar{X})(R_i - \bar{R})]}{\sqrt{\sum_{i=1}^{N}(w_i(X_i - \bar{X})^2)}\sqrt{\sum_{i=1}^{N}(w_i(R_i - \bar{R})^2)}} \tag{6}$$

A larger $r$ corresponds to a better performance. Again, more information on these weights is given in the beginning of Section 2.2.

### 2.2.4   Weighted and unweighted Earth mover's distance

The Earth mover's distance (EMD), also known as first-order Wasserstein metric $W_1$, is a metric to measure the similarity between two probability distributions of datasets $X$ and $R$ (Rubner et al., 2000). It
can be understood as the minimum amount of work needed to transform one distribution into the other. This concept is often explained using the analogy of moving piles of earth, where the EMD quantifies the cost required to move the earth from one pile to another, with the cost being proportional to the amount of earth moved and the distance it has travelled. Recently, the EMD has gained more attention for applications in climate science such as an evaluation of the performance of climate models (e.g. Vissio et
al., 2020). Here, we implement the EMD similar to Vissio et al. (2020) but for 1-dimensional distributions (i.e., to one variable at a time) and focusing on the $W_1$ metric (i.e., the EMD) only. First, we use data binning over all dimensions over which the EMD is calculated to get normalized probability mass functions $p_x(x_i)$ and $p_r(r_i)$ with $n$ bins. Here, $x_i$ and $r_i$ are the bin centers of $X$ and $R$, respectively. The bins range from the minimum to the maximum value of the data calculated over both the dataset and reference
dataset; thus, $x_i = r_i$ for all $i$. For the weighted EMD, each value only contributes with its associated weight $w$ to the bin count; for the unweighted EMD, each value contributes with equal weight. Details on the





weighting is given at the beginning of Section 2.2 With these probability mass functions, the EMD can be expressed as

$$EMD = \min_{\gamma \in \mathbb{R}_+^{n \times n}} \sum_{i,j}^{n} \gamma_{ij} |x_i - r_j| \quad \text{with} \quad \sum_{j}^{n} \gamma_{ij} = p_x(x_i); \sum_{i}^{n} \gamma_{ij} = p_r(r_j) \tag{7}$$


Here, $\gamma$ is the joint probability distribution of $x$ and $r$ (also called "optimal transport matrix") with marginals $p_x$ and $p_r$ that minimizes the transportation cost. The EMD is not sensitive to the number of bins $n$ and provides robust results even with small values of $n$ (Vissio et al., 2020; Vissio and Lucarini, 2018). The default value in ESMValTool is $n$=100, but that can be changed by the user if desired. Since 150 the EMD is a true metric in the mathematical sense, smaller values of EMD correspond to a better performance.

## 2.3 Datasets

In the following, all observational datasets used as a reference for the examples below are briefly described, listed in alphabetical order. For more details, we refer to the references given in the subsections.

### 2.3.1 Observational data

**CERES-EBAF**

The Clouds and the Earth's Radiant Energy System (CERES) Energy Balanced and Filled (EBAF) Ed4.2 dataset (Kato et al., 2018; Loeb et al., 2018) provides global monthly mean top-of-atmosphere (TOA) longwave (LW), shortwave (SW), and net radiative fluxes under clear-sky and all-sky conditions. CERES 160 instruments are on board NASA's Terra and Aqua satellites. These are used to calculate the TOA longwave (lwcre) and shortwave (swcre) cloud radiative effect as differences between the TOA all-sky and clear-sky radiative fluxes. The dataset covers the time period 2001-2022 on a global 1° x 1° grid.

**ERA5**

ERA5 is the fifth generation reanalysis of the European Centre for Medium-range Weather Forecasts 165 (ECMWF) (Hersbach et al., 2020) replacing the widely used ERA-Interim reanalysis (Dee et al., 2011).



ERA5 uses a four-dimensional variational (4D-Var) data assimilation scheme and Cycle 41r2 of the Integrated Forecasting System (IFS) (Copernicus Climate Change Service, 2017). Here, we use ERA5 data served on the Copernicus Climate Change Service Climate Data Store (CDS) that are interpolated to a horizontal resolution of 0.25° x 0.25° and in case of 3-dim variables to 37 pressure levels ranging from
1000 hPa near the surface to 1 hPa (Ecmwf, 2020).

**GPCP-SG**

The Global Precipitation Climatology Project (GPCP) is a community-based analysis of precipitation that covers the satellite era from 1979 to present. The data are produced by merging different data sources including passive microwave-based rainfall retrievals from satellites (SSMI, SSMIS), infrared rainfall
estimates from geostationary (GOES, Meteosat, GMS, MTSat) and polar-orbiting satellites (TOVS, AIRS), and surface rain gauges (Adler et al., 2018; Adler et al., 2003). Here, we use version 2.3 of GPCP-SG that provides monthly mean precipitation rates on a global 2.5° x 2.5° grid from January 1979 to present. GPCP-SG is widely used as a reference dataset for precipitation (e.g. Bock et al., 2020; Eyring et al., 2021; Hassler and Lauer, 2021; M. Nützel et al., 2023).

**HadCRUT5**

The Met Office Hadley Centre/Climatic Research Unit global surface temperature dataset HadCRUT5 contains monthly averaged near-surface temperature anomalies on a regular 5° x 5° grid from 1850 to near-present. HadCRUT5 combines sea surface temperature measurements from ships and buoys and near-surface air temperature measurements from weather stations over land. There are two versions of
HadCRUT5 available, a version representing temperature anomalies for the measurement locations ("non-infilled") and a second version for which a statistical method has been applied for a more complete data coverage ("analysis") (Morice et al., 2021). Here, we use the ensemble mean of the "analysis" version of the dataset. HadCRUT5 is widely used as a reference dataset for near-surface temperature (e.g. Eyring et al., 2021; Uribe et al., 2022).

**HadISST**

The Hadley Centre Sea Ice and Sea Surface Temperature dataset (HadISST) provides a combination of monthly globally-complete fields of SST and sea ice concentration on a 1° x 1° grid from 1870 to date.



The SST data are taken from the Met Office Marine Data Bank (MDB) with input from the International Comprehensive Ocean-Atmosphere Data Set (ICOADS) where there are no data from MDB available
(Rayner et al., 2003). For the example shown below, we use HadISST version 1.1 monthly average sea surface temperature.

**ISCCP-FH**

The International Satellite Cloud Climatology Project radiative flux profile dataset (ISCCP-FH; Zhang and Rossow (2023)) provides radiative flux profiles with a global resolution of 1°x1° at 3-hourly and
monthly intervals. ISCCP-FH data that are available over the time period July 1983 through June 2017 are based on ISCCP H-series products derived from different geostationary and polar-orbiting satellite imagers (Young et al., 2018). Here, we use the monthly means TOA clear-sky and all-sky radiative fluxes to calculate the shortwave and longwave cloud radiative effects for comparison with the models.

### 2.3.2 Model data

**CMIP6**

In this study we use data from models participating in the latest phase of the Coupled model Intercomparison Project (CMIP6, Eyring et al., 2016) for putting model deviations from observations into the context of current ESMs. For this, we use results from the "historical" simulations, for which forcings due to natural causes such as volcanic eruptions and solar variability as well as human factors such as
$CO_2$ and aerosol concentrations or land use were prescribed for the time period 1850-2014. For the examples shown in this article, we use only one ensemble member (typically the first member "r1i1p1f1") per model as the inter-model spread is typically much larger than the inter-model spread given by different ensemble members from the same model (e.g. Lauer et al., 2023). Table 1 provides an overview of the CMIP6 models used.

**Table 1** List of CMIP6 models providing data from the historical simulation that are compared with an example simulation from the EMAC model (see below) and put into the context of current ESMs. If more than one ensemble member is available, only the first ensemble member (typically "r1i1p1f1") is used.

| Model name | Institute(s) | Scientific reference(s) |
| --- | --- | --- |



| | | |
|---|---|---|
| **ACCESS-CM2** | Commonwealth Scientific and Industrial Research Organisation (CSIRO), Australian Research Council Centre of Excellence for Climate System Science (ARCCSS) | Bi et al. (2020) |
| **ACCESS-ESM1-5** | CSIRO, ARCCSS | Ziehn et al. (2020) |
| **AWI-CM-1-1-MR** | Alfred Wegener Institute, Helmholtz Centre for Polar and Marine Research (AWI), Germany | Semmler et al. (2020) |
| **AWI-ESM-1-1-LR** | AWI | Rackow et al. (2018); Sidorenko et al. (2015) |
| **BCC-CSM2-MR** | Beijing Climate Center, China | Wu et al. (2019) |
| **BCC-ESM1** | Meteorological Administration, China | Wu et al. (2020) |
| **CAMS-CSM1-0** | Chinese Academy of Meteorological Sciences (CAMS), China | Rong et al. (2018) |
| **CanESM5** | Canadian Center for Atmospheric Research (CARE), Canada | Swart et al. (2019) |
| **CanESM5-CanOE** | CARE | Swart et al. (2019) |
| **CESM2-FV2** | National Science Foundation (NSF), Department of Energy (DOE), National Center for Atmospheric Research (NCAR), USA | Danabasoglu et al. (2020) |
| **CESM2** | NSF, DOE, NCAR | Danabasoglu et al. (2020) |
| **CESM2-WACCM** | NSF, DOE, NCAR | Gettelman et al. (2019); Danabasoglu et al. (2020) |
| **CESM2-WACCM-FV2** | NSF, DOE, NCAR | Gettelman et al. (2019); Danabasoglu et al. (2020) |





| CIESM | Department of Earth System Science, Tsinghua University, China | Lin et al. (2020) |
|---|---|---|
| CNRM-CM6-1-HR | Météo-France/Centre National de Recherches Météorologiques (CNRM) and Centre Européen de Recherches et de Formation Avancée en Calcul Scientifique (CERFACS), France | Voldoire et al. (2019) |
| CNRM-ESM2-1 | CNRM, CERFACS | Séférian et al. (2019) |
| FGOALS-f3-L | CAMS | Guo et al. (2020) |
| FGOALS-g3 | CAMS | Li et al. (2020) |
| FIO-ESM-2-0 | First Institute of Oceanography, Ministry of Natural Resources (FIO), China, Qingdao National Laboratory for Marine Science and Technology (QNLM), China | Bao et al. (2020) |
| GFDL-ESM4 | National Oceanic and Atmospheric Administration (NOAA) /Geophysical Fluid Dynamics Laboratory (GFDL), USA | Dunne et al. (2020) |
| GISS-E2-1-G | National Aeronautics and Space Administration (NASA), Goddard Institute for Space Studies (GISS), USA | Rind et al. (2020) |
| GISS-E2-1-H | NASA, GISS | Rind et al. (2020) |
| HadGEM3-GC31-LL | Met Office Hadley Centre (MOHC), UK | Williams et al. (2018); Kuhlbrodt et al. (2018) |
| HadGEM3-GC31-MM | MOHC | Williams et al. (2018); Kuhlbrodt et al. (2018) |
| INM-CM4-8 | Institute for Numerical Mathematics (INM), Russian Academy of Science (RAS), Russia | Volodin et al. (2018) |
| INM-CM5-0 | INM, RAS | Volodin et al. (2017) |



| | | |
|---|---|---|
| **ISPL-CM6A-LR** | L'Institut Pierre-Simon Laplace (IPSL), France | Boucher et al. (2020) |
| **KACE-1-0-G** | National Institute of Meteorological Sciences/Korea Meteorological Administration, Climate Research Division, Republic of Korea | Lee et al. (2020) |
| **MCM-UA-1-0** | Department of Geosciences, University of Arizona, USA | Delworth et al. (2002) |
| **MIROC6** | Japan Agency for Marine-Earth Science and Technology (JAMSTEC), Atmosphere and Ocean Research Institute (AORI), University of Tokyo, and National Institute for Environmental Studies (NIES), Japan | Tatebe et al. (2019) |
| **MIROC-ES2L** | JAMSTEC, AORI, NIES | Hajima et al. (2020) |
| **MPI-ESM-1-2-HAM** | HAMMOZ-Consortium: ETH Zurich, Switzerland; Max Planck Institut für Meteorologie (MPIM), Germany; Forschungszentrum Jülich, Germany; University of Oxford, UK; Finnish Meteorological Institute, Finland; Leibniz Institute for Tropospheric Research, Germany; Center for Climate Systems Modeling (C2SM) at ETH Zurich, Switzerland | Mauritsen et al. (2019) |
| **MPI-ESM1-2-HR** | MPIM | Muller et al. (2018) |
| **MPI-ESM1-2-LR** | MPIM | Mauritsen et al. (2019) |
| **MRI-ESM2-0** | Meteorological Research Institute (MRI), Japan | Yukimoto et al. (2019) |
| **NESM3** | Nanjing University of Information Science and Technology, China | Cao et al. (2018) |
| **NorESM2-LM** | NorESM Climate modeling Consortium (NCC), Norway | Seland et al. (2020) |
| **NorESM2-MM** | NCC | Seland et al. (2020) |



| SAM0-UNICON* | Seoul National University, Republic of Korea | Park et al. (2019) |
| --- | --- | --- |
| UKESM1-0-LL | MOHC | Sellar et al. (2019) |

## EMAC

The ECHAM/MESSy Atmospheric Chemistry (EMAC) model is a chemistry-climate model (Jöckel et
al., 2010) that has been widely used for various studies in atmospheric sciences including, for instance,
tropospheric and stratospheric ozone (e.g. Dietmüller et al., 2021; Mertens et al., 2021), climate impact
of contrails and emissions from aviation (e.g. Frömming et al., 2021; Matthes et al., 2021) and the effects
of transport on atmosphere and climate (e.g. Hendricks et al., 2018; Righi et al., 2015). EMAC uses the
second version of the Modular Earth Submodel System (MESSy2) to link submodels for various physical
and chemical processes to the host model. Here, the 5th generation of the European Centre Hamburg
general circulation model (ECHAM5; Roeckner et al. (2006)) is used as host model.

In this study, we use an EMAC simulation with deliberately erroneous prescribed sea surface temperatures
(SSTs) to showcase application of the new ESMValTool features to monitor and benchmark model
simulations during the model development phase with results from established global climate models.
While a comparison of results from coupled historical CMIP6 simulations with an AMIP-style simulation
in which SSTs and sea ice concentrations are prescribed from observations is of course not completely
fair for a real model benchmarking or monitoring of a simulation, this approach allows us to showcase
the new ESMValTool features with a simulation in which something goes wrong after a few simulation
years. For this, the SST fields are set to zonally averaged monthly values of the observed global average
SST after the first five years of model simulation (see Figure 1). Such an error does not necessarily show
up in time series of global mean near-surface temperature but can be identified when using other metrics.



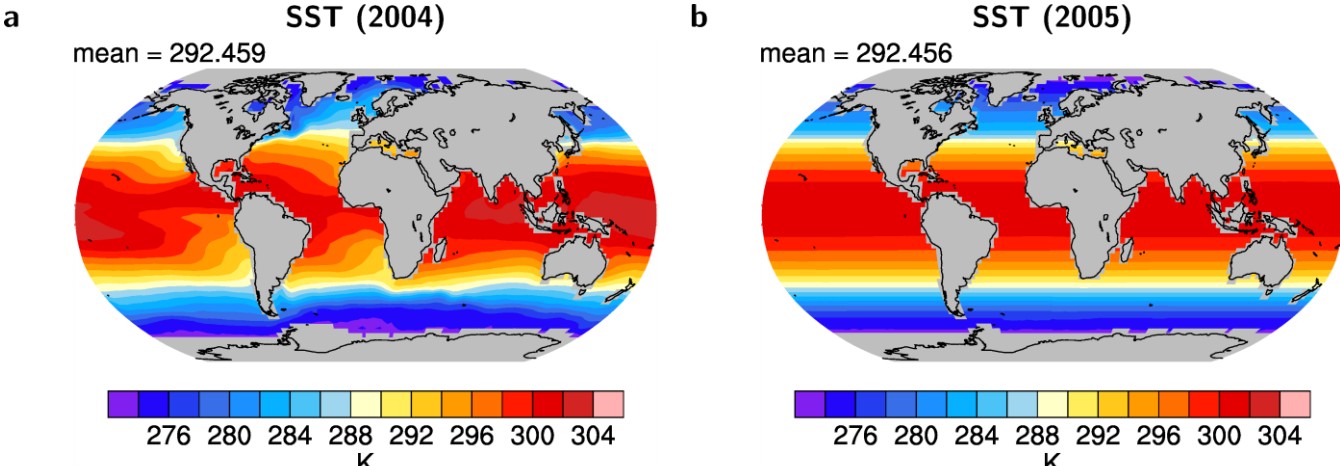

**Figure 1** Annual mean of the prescribed sea surface temperatures (SSTs) for the EMAC simulation (a) before (year 2004) and (b) after (year 2005) the deliberately introduced "error".

## 3  Monitoring and benchmarking of ESM simulations

In the following, we show examples of how the new ESMValTool capabilities can be used to monitor and benchmark model simulations to detect problems during runtime and to assess whether the performance of a model simulation is within the range of what could be expected from current state-of-the-art ESMs (here: CMIP6). The variables and reference datasets used in the examples are listed in Table 2.

**Table 2** Variables and reference datasets used.

| Variable | Description | Reference dataset(s) |
|---|---|---|
| **tas** | near-surface air temperature (K) | HadCRUT5, ERA5 |
| **tas_land** | same as tas but over land grid cells only (K) | HadCRUT5, ERA5 |
| **sst** | sea surface temperature (K) | HadISST, ERA5 |
| **pr** | precipitation (mm day$^{-1}$) | GPCP-SG, ERA5 |
| **psl** | air pressure at sea level (Pa) | ERA5 |
| **ta** | air temperature (K) | ERA5 |
| **rlut** | TOA outgoing longwave radiation (W m$^{-2}$) | CERES-EBAF, ISCCP-FH |
| **rsut** | TOA outgoing shortwave radiation (W m$^{-2}$) | CERES-EBAF, ISCCP-FH |



| lwcre | TOA longwave cloud radiative effect (W m$^{-2}$) | CERES-EBAF, ISCCP-FH |
|---|---|---|
| swcre | TOA shortwave cloud radiative effect (W m$^{-2}$) | CERES-EBAF, ISCCP-FH |

## 3.1  Time series

Time series of climate relevant quantities or their anomalies relative to a given reference period averaged over a specific region or the entire globe are a common approach to evaluate model results with one or several reference datasets (e.g., Bock et al., 2020; Yazdandoost et al., 2021; Wang et al., 2023). As an example, Figure 2a shows a time series of global average anomalies in near-surface temperature. In addition to the EMAC model results (red line) and the observational reference data from HadCRUT5 (black line), also the CMIP6 results (Table 1) are shown as thin gray lines. The figure shows that the first five years of the EMAC simulation are rather at the high end of the CMIP6 results with the temperature anomalies frequently exceeding the 90 % percentile of the CMIP6 results. In the beginning of the year 2005, there is the sudden temperature drop when the deliberate error in the SST fields is introduced resulting in the EMAC simulation being at the low end of the CMIP6 range with temperature anomalies frequently being below the CMIP6 10 % percentile. Figure 2b shows a time series of the global average (area weighted) root mean square errors in simulated near-surface temperature from EMAC (red line). The 10 % and 90 % percentile range of the RMSE values from the individual CMIP6 models is shown as light blue shading. The "error" in the geographical distribution of the sea surface temperatures introduced in 2005 is not obvious in this time series as the performance of this EMAC simulation is within the range of what could be expected from a coupled CMIP6 model. This shows that monitoring of model simulations typically requires assessing several variables.



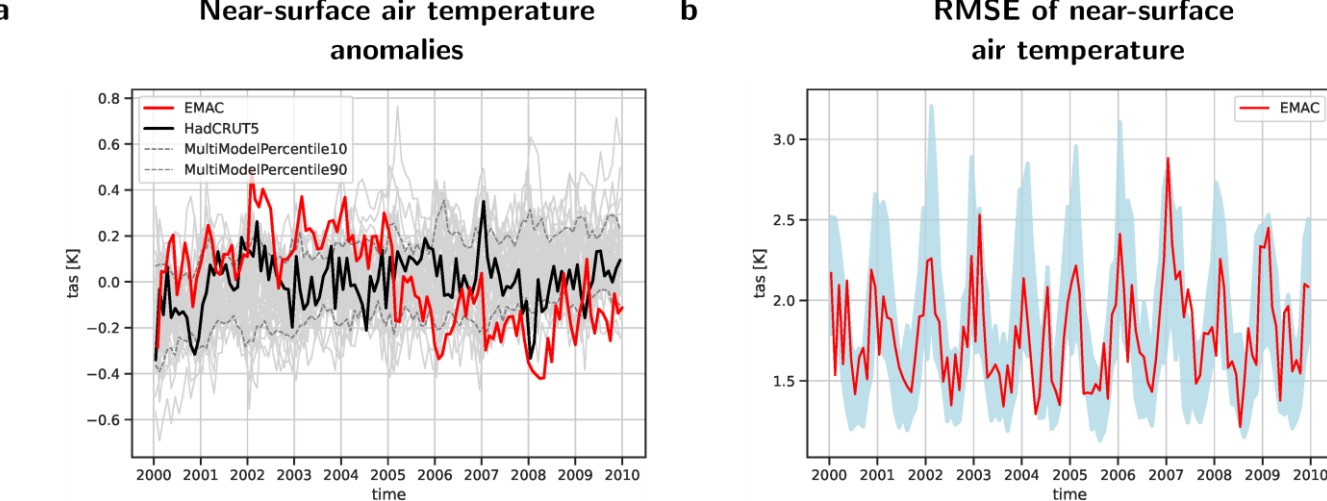


**Figure 2** (a) Time series from 2000 through 2009 of global average monthly mean temperature anomalies (reference period 2000-2009) of the near-surface temperature in K from a simulation of EMAC (red) and the reference dataset HadCRUT5 (black). The thin gray lines show 43 individual CMIP6 models used for comparison, the dashed gray lines show the 10 % and 90 % percentiles of these CMIP6 models. (b) Same as (a) but for area-weighted RMSE of the near-surface air temperature. The light blue shading shows the range of the 10 % to 90 % percentiles of RMSE values from the ensemble of 43 CMIP6 models used for comparison.

### 3.2 Diurnal and seasonal cycle

A further commonly used metric for model evaluation is the comparison of the seasonal cycle of a specific

variable, calculated for the whole globe or again for a pre-defined region. Figure 3a shows the multi-year global mean seasonal cycle of near-surface air temperature for a suite of CMIP6 models, the HadCRUT5 observations, and the specifically created EMAC simulation that has been described in Section 2.3.2. The CMIP6 model simulations and the HadCRUT5 data are averaged over the time period 2000-2009, whereas the EMAC simulation is split in the two five-year periods without and with the erroneous SSTs,

2000-2004 (red line) and 2005-2009 (dark blue line) respectively. Similar to Figure 2a, also Figure 3a indicates the 10 % and 90 % percentile range with gray dashed lines. Both five-year means of the EMAC simulation are well within the CMIP6 10 % and 90 % percentile range throughout the whole year, but the EMAC simulation period with the correct SSTs is slightly closer to the HadCRUT5 data than the simulation period with the erroneous SSTs. While this is positively noted, it itself is not a clear indication

that a problem occurred with the latter five-year period of the EMAC simulation. Figure 3b shows then





the area-weighted RMSE values for the global mean seasonal cycle of near-surface air temperature. The blue shading depicts the 10 % to 90 % percentile range of the CMIP6 models used for the comparison. The earlier five-year period of the EMAC simulation (2000-2004, red line) is in most months below the blue shaded area, which means that with correct SST fields the example EMAC simulation can reproduce
the seasonal cycle of near-surface temperature better than most CMIP6 models (smaller RMSE = better performance). With the erroneous SSTs, however, the RMSE values for the annual cycle become larger, which means that the agreement of the seasonal cycle of near-surface air temperature with the reference dataset decreased for that period of the EMAC simulation. They are still located within the blue-shaded area, but agreement is less good than for the earlier period (red line). Again, this metric alone would not
allow the clear detection of a faulty simulation, but it would be clear that in "normal" simulations EMAC is performing clearly better than most CMIP6 models when looking at the RMSE of near-surface air temperature, and the clear decrease in performance could be an indicator that something might be problematic with a new simulation.

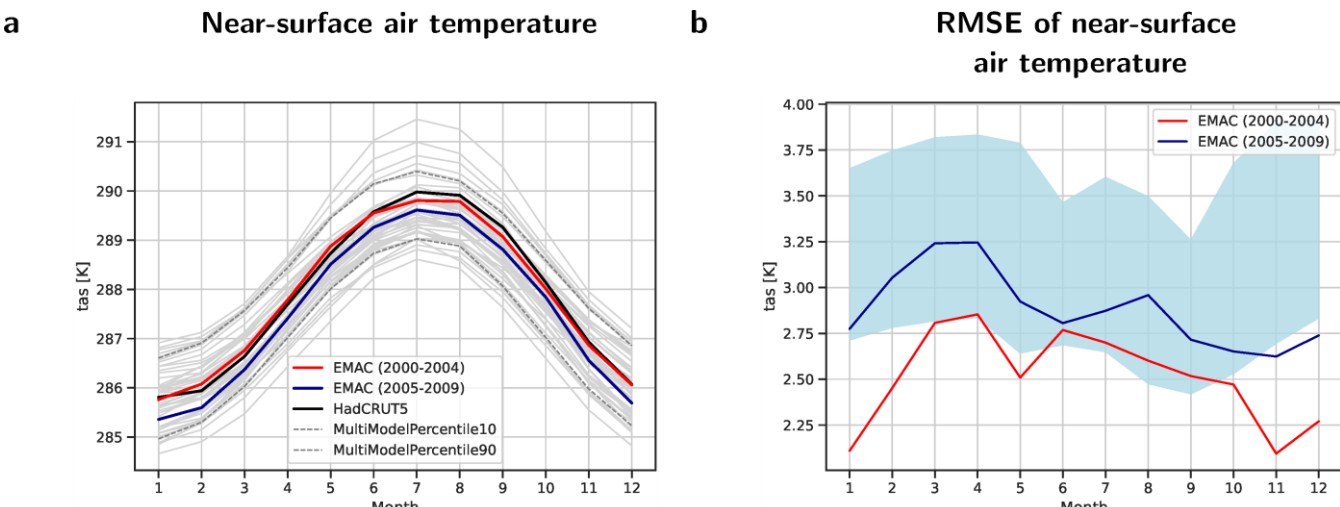

**Figure 3** (a) Multi-year global mean of the seasonal cycle of near-surface air temperature in K from a simulation of EMAC averaged over the time period 2000-2004 (red) and 2005-2009 (dark blue) and the reference dataset HadCRUT5 (2000-2009, black). The thin gray lines show 43 individual CMIP6 models (2000-2009) used for comparison, the dashed gray lines show the 10 % and 90 % percentiles of these CMIP6 models. (b) Same as (a) but for area-weighted RMSE of near-surface temperature. The light blue



shading shows the range of the 10 % to 90 % percentiles of RMSE values from the ensemble of 43 CMIP6 models used for comparison.

A further capability implemented in ESMValTool is to intercompare the diurnal cycle of a variable, for example precipitation (see Figure 4). The basic structure of the graphs is identical to Figure 3 regarding the shown EMAC simulation, the CMIP6 simulations and their spread. The example results show, however, the precipitation averaged only over the tropical ocean instead of a global mean and averaged over only two years (2004-2005). ERA5 has been used as reference dataset. Both years of the EMAC simulation show a reduced amplitude of the average diurnal cycle of precipitation over the tropical ocean compared to ERA5 and most of the CMIP6 ensemble (Figure 4a), with 2004 (from the "correct" period) being even further away from the reference compared to 2005 (from the erroneous period). Figure 4b shows the RMSE of the diurnal cycle of precipitation over the tropical ocean. The blue shaded region indicates again the 10 % to 90 % percentile range of the CMIP6 models. The year 2004 of the EMAC simulation is fully enclosed by the CMIP6 percentile range, whereas 2005 is for most hours of the day above the CMIP6 percentile range. This reversal of which EMAC simulation year performs better compared with ERA5 suggests, that some kind of error compensation takes place when calculating the mean values (Figure 4a), while this is not the case when calculating the RMSE value at each grid cell for a given time of the day and then averaging afterwards (Figure 4b). Similar to the metric shown in Figure 3, the comparison of the diurnal cycle of precipitation alone might not be able to correctly identify erroneous simulations, but also this metric could give an indication that something might not be correct with a new simulation, if it is possible to compare it to a "baseline" simulation of the same model that has been labeled as "correct".



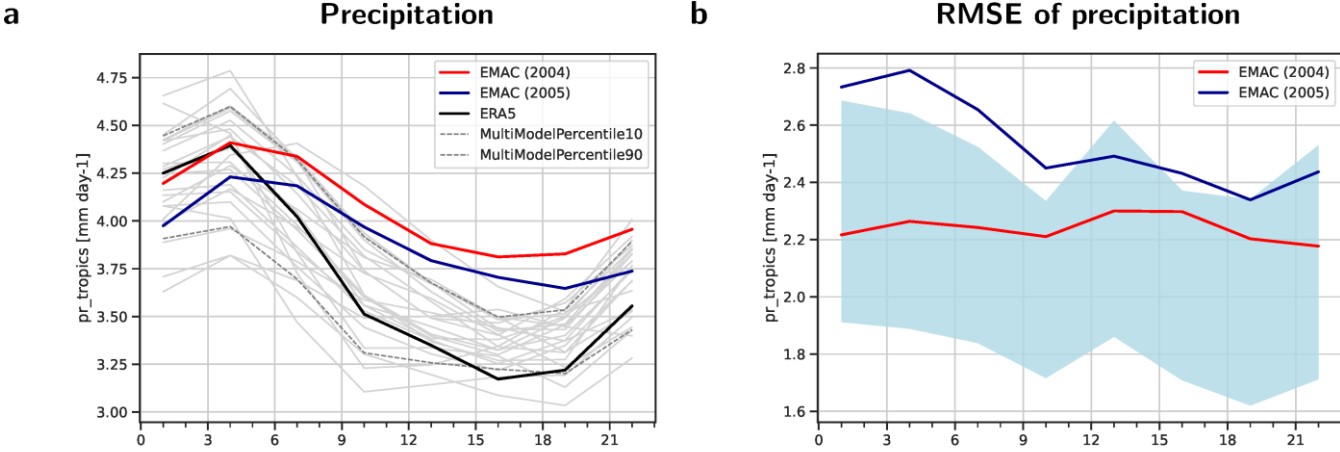

**Figure 4** Annual mean diurnal cycle of precipitation averaged over the tropical ocean (ocean grid cells in the latitude belt 30° S to 30° N) from a simulation of EMAC averaged over the year 2004 (red) and over 2005 (dark blue) compared with ERA5 data (2004-2005, black). The thin gray lines show 22 individual CMIP6 models used for comparison (2004-2005), the dashed gray lines show the 10 % and 90 % percentiles of these CMIP6 models. (b) Same as (a) but for area-weighted RMSE of precipitation. The light blue shading shows the range of the 10 % to 90 % percentiles of RMSE values from the ensemble of 22 CMIP6 models used for comparison.

## 3.3 Geographical distribution

Figure 5 shows an example of how the RMSE of the time series of monthly mean precipitation at each grid cell from a given simulation can be compared with the range of RMSE values from the CMIP6 models. As a reference, GPCP-SG data are used (Sect. 2.3.1). The stippled grid cells denote areas at which the RMSE value of the given simulation is below the 90 % percentile of RMSE values from the CMIP6 models. This threshold can be set depending on what is considered OK during model development or

model benchmarking, allowing to focus on the non-stippled areas showing larger deviations. Figure 5a shows the RMSE of the precipitation time series of the EMAC simulation for the period 2000–2004. In this figure, non-stippled areas are mainly found in the tropical East Pacific and Indian Ocean highlighting the regions that show larger RMSE values than most of the CMIP6 models and that might need further investigation during model development, or that perform worse than what could be expected from a state-

of-the-art model. As a result of the deliberately introduced error in the geographical SST distribution in 2005, these areas are much larger in the second half (2005–2009) of the EMAC simulation (see Figure 5b) and now cover most of the tropical oceans. When applied to the monitoring of a running simulation,





this increase in areas performing less well than the majority of CMIP6 models can be a first indication of problems related to deep convection, which requires further investigation.

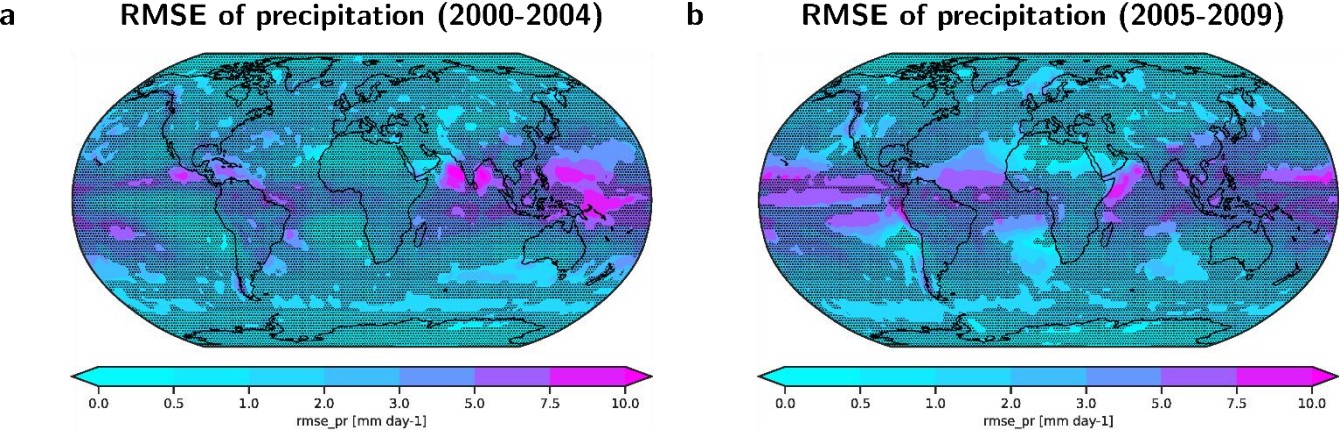


**Figure 5** 5-year annual mean area-weighted RMSE of the precipitation rate in mm day$^{-1}$ from a simulation of EMAC compared with GPCP-SG data. (a) Average over the time period 2000-2004, (b) average over the time period 2005-2009. The stippled areas mask grid cells where the RMSE is smaller than the 90 % percentile of RMSE values from an ensemble of 39 CMIP6 models.

## 3.4 Zonal averages


For 3-dimensional variables such as air temperature, a comparison of zonally averaged fields with reference data is an easy and common way to evaluate a model simulation. For this, the absolute or relative bias can be used as a measure of how well the model simulation reproduces the reference data. In Figure 6, the absolute bias of the EMAC example simulation compared with ERA5 data for the zonally averaged 3-dimensional air temperature is shown. Here, the stippling indicates that the absolute value of the bias $|bias|$ is smaller than the maximum of the absolute 10 % and the absolute 90 % percentiles, $|p10|$ and $|p90|$, respectively, of the bias values from the CMIP6 ensemble for this grid cell. By using the criteria $|BIAS| \leq max(|p10|, |p90|)$, positive and negative bias values are given the same importance when assessing the model performance. Depending on the aim of the model development and the percentiles



selected for this comparison, all non-stippled bias values outside of this range can be regarded as below par performance and might require further investigation and possibly continued model improvements or model tuning. When monitoring a running simulation, the strong increase in the grid cells that are marked as below-average performance between the first (Figure 6a) and the second simulation time period (Figure





6b) is a first hint that there might be an unexpected problem in the simulation that occurred during run-
time.

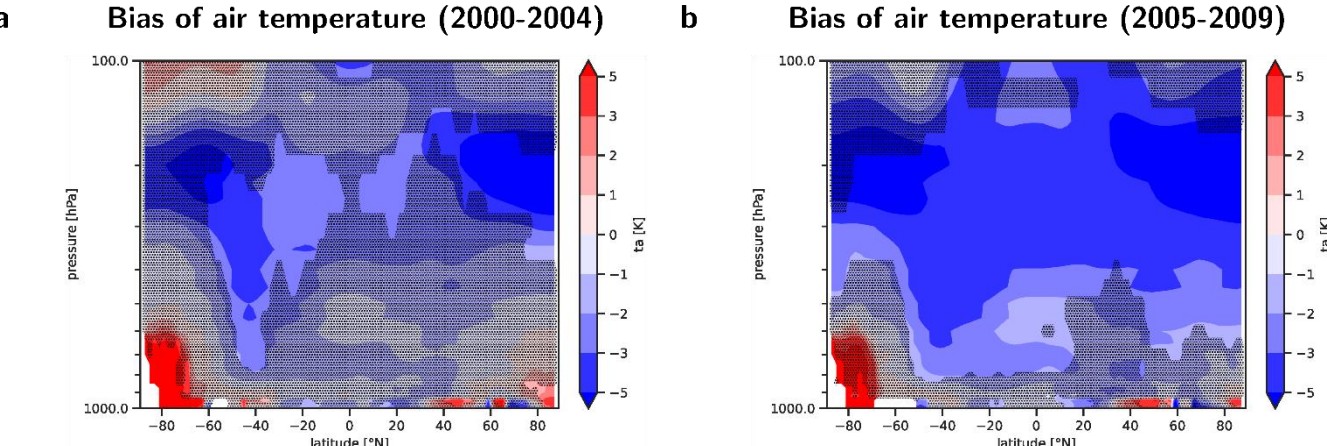

**Figure 6** 5-year annual mean bias of the zonally averaged temperature in K from a historical simulation of the EMAC model compared with ERA5 reanalysis data. (a) Average over the time period 2000-2004, (b) average over the time period 2005-2009. The stippled areas mask grid cells where the absolute BIAS
($|\mathbf{BIAS}|$) is smaller than the maximum of the absolute 10 % ($|\mathbf{p10}|$) and the absolute 90 % ($|\mathbf{p90}|$) percentiles from an ensemble of 38 CMIP6 models, i.e. $|\mathbf{BIAS}| \leq \mathbf{max}(|\mathbf{p10}|, |\mathbf{p90}|)$.

## 3.5 Box plots

The summary plots for different variables as shown in Figure 7 offer the possibility to quickly get a first overview on model performance. It can either be used as a starting point for a more in-depth evaluation
of individual variables or climate parameters with observations, or as one possible summary of overall model performance. For every diagnostic field considered, model performance is compared to one reference dataset (see Table 2, first dataset), and the quality of the simulation is summarized in a single number such as RMSD (Figure 7a,b), Pearson's correlation coefficient (Figure 7c,d) or EMD (Figure 7e,f) computed over the time averaged global maps.

By simultaneously assessing a number of different performance indices, the general model improvements can then be quantified and compared with the CMIP6 ensemble. In our example EMAC simulation, the SSTs are prescribed; thus, we see a significantly better performance in SST than the CMIP ensemble of coupled (historical) simulations especially regarding the RMSE (Figure 7a) and the Pearson's correlation coefficient (Figure 7c). For the other variables, the EMAC example shows often a slightly worse



performance than 75 % percentile of the CMIP6 models, but mostly lies still in the range of the CMIP6 models. This changes when we look at the second time period (Figure 7b), where we can see a significant decrease of model performance regarding RMSE for all variables. Furthermore, it can be seen that the decrease of performance in the second time period is most prominent for the SSTs especially in the RMSE and correlation pattern values (Figure 7b,d). This is a clear hint that detailed diagnostics for this variable

(e.g. see Figure 2) would be helpful in order to quickly identify the error in the simulation.





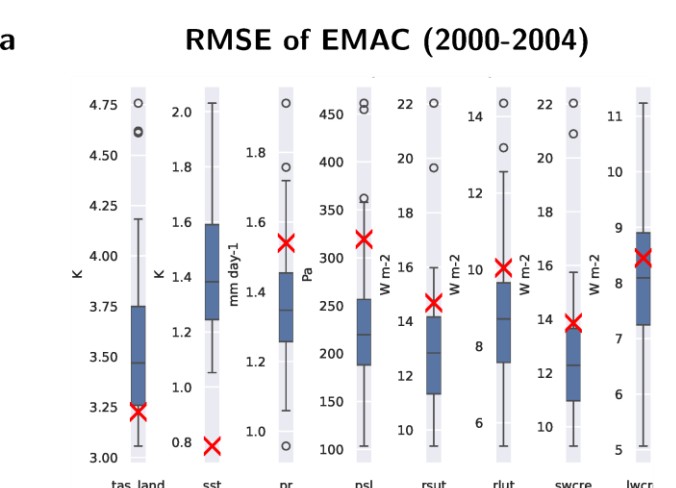

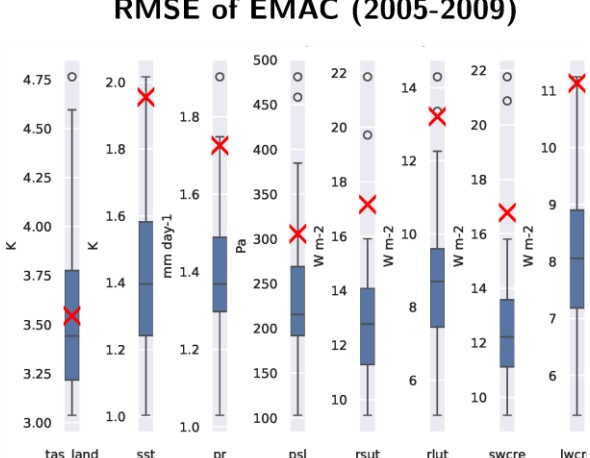

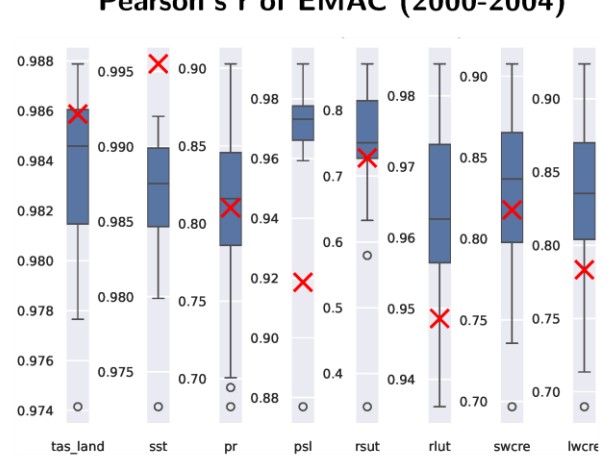

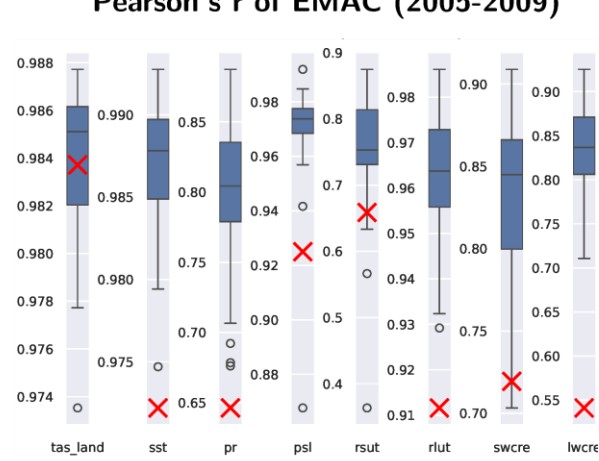

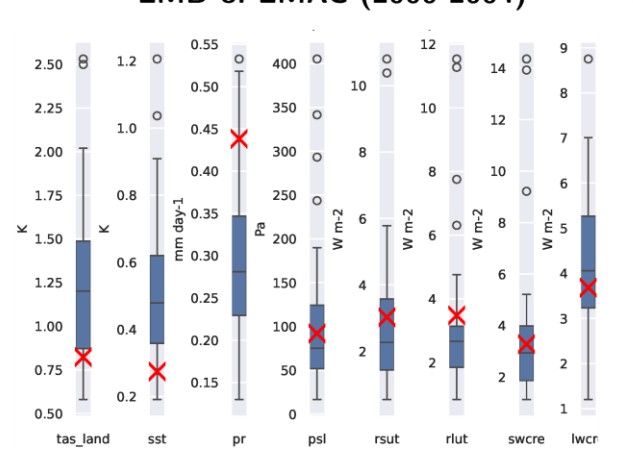

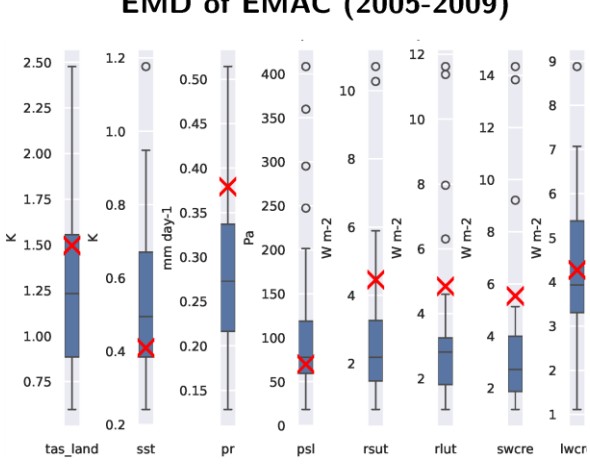



**Figure 7** (a, b) Global area-weighted RMSE (smaller=better), (c, d) weighted Pearson's correlation coefficient (higher=better) and (e, f) weighted Earth mover's distance (smaller=better) of the geographical pattern of 5-year means of different variables from a simulation of EMAC (red cross) in comparison to the CMIP6 ensemble (boxplot). The left column shows the results for the time period 2000-2004, the right column for 2005-2009. Reference datasets for calculating the three metrics are: near-surface temperature (tas): HadCRUT5, surface temperature (ts): HadISST, precipitation (pr): GPCP-SG, air pressure at sea level (psl): ERA5, shortwave (rsut) longwave (rlut) radiative fluxes at TOA and shortwave (swcre) and longwave (lwcre) cloud radiative effects: CERES-EBAF. Each box indicates the range from the first quartile to the third quartile, the vertical lines show the median, and the whiskers the minimum and maximum values, excluding the outliers. Outliers are defined as being outside 1.5 times the interquartile range.

## 3.6 Portrait diagram

Portrait diagrams (Gleckler et al., 2008) can be used to visualize model performance across different variables relative to one or multiple reference datasets. Unlike box plots, portrait diagrams show the performance of each model individually; thus, they provide a convenient way to benchmark each element in an ensemble of models. Figure 8 shows an example of a portrait diagram for the same set of variables as used in the box plots (see Figure 7). The horizontal axis shows the different models (left: CMIP6 models; right: the EMAC simulation for two different time periods) and the vertical axis the different variables. The colors correspond to the relative RMSE (relative to the median RMSE across all models) of the different models and variables, where red corresponds to a higher RMSE (= worse performance), and blue to a lower RMSE (= better performance) than the median. For variables where the box is split into two triangles, an alternative dataset is provided in the lower right triangle (see Table 2 for an overview of variables and reference datasets used). The effect of the deliberately introduced error in the EMAC simulation is clearly visible on the right side of the portrait diagram: as expected, the wrong SST pattern starting in 2005 leads to a sharp decline in the relative RMSE in SST from dark blue colors (i.e., very good performance) in 2000–2004 to dark red colors (i.e., very bad performance) in 2005–2009. However, the error is not only visible in the SST: across all variables, the later period (2005–2009) of the EMAC simulation shows a higher relative RMSE (i.e., worse performance) than the corresponding early period (2000–2004). In addition to RMSE, also the metrics EMD or Pearson correlation coefficient could be used (see Sect. 2.2).





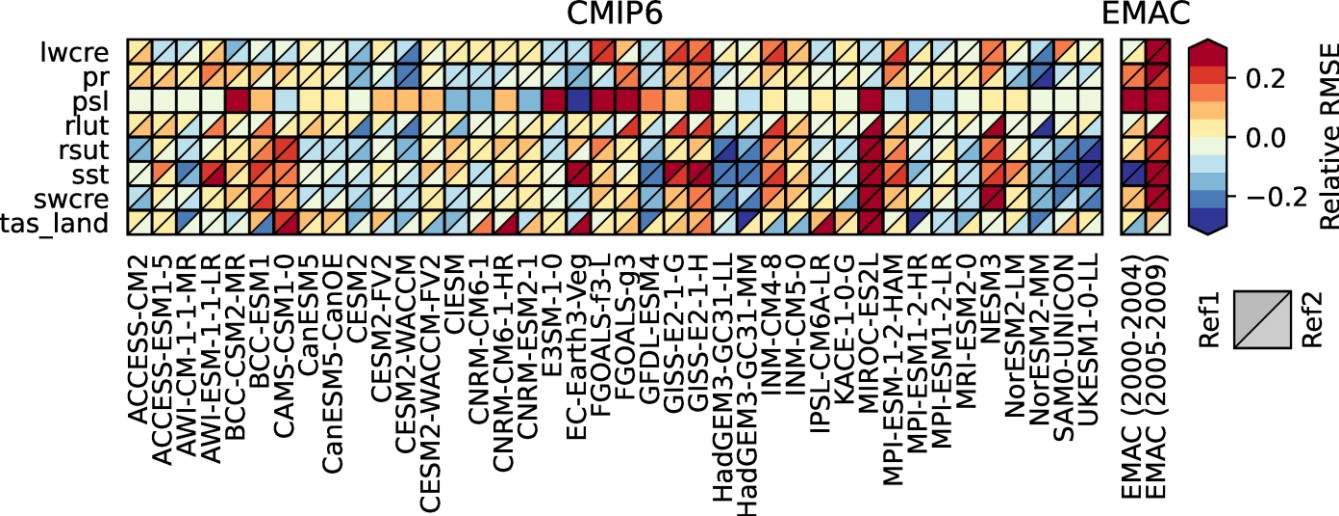

**Figure 8** Portrait diagram showing the relative space-time root-mean-square error (RMSE) calculated from the seasonal cycle of the datasets. The seasonal cycle is averaged over the years 2000-2009 (CMIP6 models) and over the time periods 2000-2004 and 2005-2009 for the EMAC simulation. The figure shows the relative performance with blue shading indicating a better, and red shading indicating a worse performance than the median RMSE of all models. The lower right triangle shows the relative RMSE with respect to the reference dataset (Ref1), the upper left triangle with respect to an alternative reference dataset (Ref2). Using RMSE as a metric (as shown) gives a portrait diagram similar to Gleckler et al. (2008). Other metrics are available.

## 4    Summary and discussion

In this paper, we introduce the newly extended capability of the Earth System Model Evaluation Tool to benchmark and monitor climate model simulations across a wide range of different Earth system components. The new framework allows to put common performance metrics calculated for a given model simulation into the context of results from an ensemble of state-of-the-art climate models such as the ones participating in the Coupled Model Intercomparison Project Phase 6. Putting the performance of a model simulation into such a context allows to quickly assess whether, for instance, the values obtained for metrics such as bias or pattern correlation for a variable are within the typical range of model errors or might need further, more detailed investigation. This is particularly of help during model development or when monitoring a simulation to identify possible problems already during run-time as this allows a large number of variables to be assessed without the need for detailed expert knowledge on each single



quantity. This is also of help when automatizing monitoring of running simulations. For this, the numerical output of ESMValTool in the form of netCDF files could be used to summarize the results from the different metrics with a e.g. dash board displayed on a web site showing a green, yellow or red traffic light for each quantity tested depending on the results. The percentiles for the metric obtained from the model ensemble used for comparison can be used as thresholds to flag quantities that are outside the range of typical model errors and thus in need of further inspection. A possible application for these new model benchmarking and monitoring capabilities of ESMValTool would be the assessment of new model simulations during the preparation phase for CMIP7.

As shown in Section 3 particularly for model development, these metrics are most effective if there are already results from a well-tuned, well-understood "baseline" simulation of the same model available. With the results for this "baseline" simulation being known the evaluation and benchmarking of a new simulation can be done already quite effectively with few simulation years since the deviation from the "baseline" become apparent quickly for many relevant atmospheric variables. For the examples shown in this paper, for instance, we found that five model years are usually already sufficient for this kind of first assessment.

The possibility to use wildcards in recipes when specifying the model datasets (available since ESMValTool version 2.8.0) used to provide context for comparison in combination with the feature to download any data that are missing locally but that are available on the ESGF automatically (available since ESMValTool version 2.4.0) makes application of ESMValTool for model benchmarking and monitoring very easy and user friendly. Examples of how to use the new capabilities of ESMValTool for benchmarking and monitoring include time series, seasonal and diurnal cycles as well as map plots, box plots and portrait diagrams for any 2-dimensional variable including individual levels or, for instance, zonally averages of 3-dimensional variables that can be shown as latitude-height plots.

The benchmarking and monitoring diagnostics introduced in this paper currently support absolute and relative bias, Pearson's correlation coefficient, root mean square error and Earth mover's distance as metrics. All of these metrics can be calculated as unweighted or weighted, e.g. by using the area size of the grid cells as weights. As all of these basic metrics are implemented in the form of a generic

preprocessing function of ESMValTool, adding new metrics is straightforward and new metrics can then
be used by all diagnostics with little to no additional effort.

## 5    Code availability Statement

ESMValTool v2 is released under the Apache License, VERSION 2.0. The latest release of ESMValTool
v2 is publicly available on Zenodo at https://doi.org/10.5281/zenodo.3387139. The source code of the
ESMValCore package, which is installed as a dependency of ESMValTool v2, is also publicly available
on Zenodo at https://doi.org/10.5281/zenodo.3401363. ESMValTool and ESMValCore are developed on
the GitHub repositories available at https://github.com/ESMValGroup.

## 6    Data Availability Statement

CMIP6 data are available freely and publicly from the Earth System Grid Federation (ESGF) and can be
retrieved by ESMValTool automatically by setting the configuration option 'search_esgf' to
'when_missing' or 'always'. All observations/reanalysis data used are described in Sect. 2.3.1. The
observational/reanalysis datasets are not distributed with ESMValTool that is restricted to the code as
open source software, but ESMValTool provides a collection of scripts with downloading and processing
instructions to recreate all observational/reanalysis datasets used in this publication. The EMAC data used
as an example in this study are available on Zenodo at http://doi.org/10.5281/zenodo.11198445.

## 7    Author contributions

AL and BH developed the concept for this work. AL, LB, LR and MS contributed to coding the
ESMValTool extensions presented, PJ designed and performed the EMAC simulation used as an
example. All authors contributed to the writing and editing of the paper.

## 8    Competing interests

One author is a member of the editorial board of the journal Geoscientific Model Development.



# 9    Acknowledgements

The development of ESMValTool is supported by several projects. The diagnostic development of ESMValTool v2 for this paper received funding from the European Union's Horizon 2020 research and innovation programme under Grant Agreement No. 101003536 (ESM2025 – Earth System Models for the Future) and the European Research Council (ERC) Synergy Grant "Understanding and Modeling the Earth System with Machine Learning (USMILE) under the Horizon 2020 research and innovation programme (Grant agreement No. 855187). We acknowledge the World Climate Research Program's (WCRP's) Working Group on Coupled Modelling (WGCM), which is responsible for CMIP, and we thank the climate modelling groups for producing and making available their model output in the framework of ESGF. The CMIP data of this study were replicated and made available for this study by the DKRZ. This work used resources of the Deutsches Klimarechenzentrum (DKRZ) granted by its Scientific Steering Committee (WLA) under project IDs bd0854, id0853 and bd1179.

This manuscript contains modified Copernicus Climate Change Service (2017) information with ERA5 data retrieved from the Climate Data Store (neither the European Commission nor ECMWF is responsible for any use that may be made of the Copernicus Information or Data it contains). ECMWF does not accept any liability whatsoever for any error or omission in the data, their availability, or for any loss or damage arising from their use.

CERES-EBAF data were obtained from the NASA Langley Research Center Atmospheric Science Data Center.

Global Precipitation Climatology Project (GPCP) Monthly Analysis Product data used are provided by the NOAA PSL, Boulder, Colorado, USA, downloaded from their website at https://psl.noaa.gov (last access 9 May 2023).

HadCRUT5 data were obtained from http://www.metoffice.gov.uk/hadobs/hadcrut5 on 6 Dec 2021 and are © British Crown Copyright, Met Office 2020, provided under an Open Government License, http://www.nationalarchives.gov.uk/doc/open-government-licence/version/3/.

HadISST v1.1 data were obtained from https://www.metoffice.gov.uk/hadobs/hadisst/ (last access 20 January 2023) and are © British Crown Copyright, Met Office, 2007, provided under a Non-Commercial



Government           Licence           http://www.nationalarchives.gov.uk/doc/non-commercial-government-licence/version/2/.

The ISCCP-FH Radiative Flux Profile Product used in this study was developed by Y. Zhang and W. Rossow and obtained from the NOAA National Centers for Environmental Information (NCEI) (https://www.ncei.noaa.gov).

We would like to thank Franziska Winterstein (DLR) for helpful comments on the manuscript.

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
