# Peer review of "Monitoring and benchmarking Earth system model simulations with ESMValTool v2.12.0"

_EGUsphere, 2024_

## Author Comment (AC1)

**Response to Reviewers**

**Geoscientific Model Development**

| | |
|---|---|
| Manuscript: | egusphere-2024-1518 |
| Title: | Monitoring and benchmarking Earth system model simulations with ESMValTool v2.12.0 |
| Authors: | Axel Lauer, Lisa Bock, Birgit Hassler, Patrick Jöckel, Lukas Ruhe, Manuel Schlund |
| Date: | 19 September 2024 |

*We thank the two reviewers for their helpful comments. In this document, we answer each point raised by the reviewers. The original reviewers' comments are given in **black**, our answers in **blue**. A "track changes" version of the revised manuscript highlighting all changes is available.*

**Reviewer #1**

This paper is documenting the recent updates of one of climate and Earth system model evaluation software package, named ESMValTool. Considering the popularity of the tool, it is important to keep the capabilities updated and well-documented as a community resource. I think the paper is well organized. I only have a few minor comments as follows.

We thank Reviewer #1 for providing helpful comments to improve the manuscript.

I wonder if authors could clarify more explicitly what are the new capabilities in this specific version of the tool, compared to the previous version with the published paper. Are those metrics in section 2.2 all new metrics that were not available in the previous version?

In the previous version, only 'bias' was available as a preprocessing function that can be applied to an ensemble of models. Correlation and RMSE were only calculated within selected diagnostics, which did not allow generic application to arbitrary quantities, dimensions (time, longitude latitude, level), geographical regions, etc. A visual comparison of the metric results among an ensemble of models was therefore not possible with previous versions. The metric Earth mover's distance has been newly added to ESMValTool v2.12.0. We made this clearer by adding the following paragraph to section 2.2:

"The metrics have been implemented as generic preprocessing functions that are newly available in v2.12.0. In contrast to previously available diagnostic-specific implementations of such metrics, the preprocessing functions can be applied to ensembles of models and arbitrary variables and dimensions providing the flexibility needed for the new benchmarking and monitoring capabilities of ESMValTool described here."

Line 44 to 47 "For this, for example results from the Coupled Model Intercomparison Phase 5 and 6 (Eyring et al., 2016; Taylor et al., 2012) can be used to get an overview of which biases can be

considered "acceptable for now", and which would need more attention and more detailed analysis and comparisons with observations.": As there have been several tools being developed for such purposes, I wonder if it would be beneficial to provide a few references as examples: e.g., PCMDI Metrics Package (Lee et al., 2024, GMD), ILAMB (with proper reference), etc.

The reviewer has a good point. We added references to PCMDI Metric Package and ILAMB to the introduction:

"A number of software tools for model evaluation has been developed over the recent years. Examples include, for instance, the PCMDI Metrics Package (PMP, Lee et al. (2024)), the International Land Model Benchmarking (ILAMB) system (Collier et al., 2018), or the Earth System Model Evaluation Tool (ESMValTool, Righi et al. (2020))."

Line 88 "For all metrics, an unweighted and weighted version exists" and sections 2.2.2 through 2.2.4: I wonder what the rationale was to include unweighted metrics. While it is fair to include both methods as options, I think weighted metrics might better considered for the "default" method. Is there any practical use case of unweighted metrics?

We agree with the reviewer that applications of weighted metrics are by far the most common use case. We implemented also the option to calculate unweighted metrics for example for application to station data, for which individual model grid cells are selected that contain the measurement station. In this case, weighting with the gridbox area instead of giving each station pixel the same weight might distort results. We added the following sentence to section 2.2:

"While the weighted version is the preferred option for most use cases, an unweighted option is available for cases where weighing with the gridbox area might distort the results. Examples of such cases include, for instance, extracting individual model grid cells containing a measurement station and giving the same weight to each station, independent of the model gridbox area."

Line 149 "The default value in ESMValTool is n=100": Does the bin distributed equally and sized evenly by min/max of the PDF? I guess this might be the case but it won't be harmful to clarify it.

Yes, the reviewer is right, the bins are distributed automatically. We clarified this by adding "equally sized" to the corresponding sentence.

Line 155 "2.3.1 Observation datasets": I think the word "reference" might better inclusively represent datasets listed in the section. Some reanalysis datasets were discussed along, but often they are preferred to be differentiated from instrument-based observation.

Agreed, we changed "observational data" to "reference data" as suggested.

Line 170 "(Ecmwf, 2000)": Please capitalize all letters for ECMWF.

Thanks for spotting this. This is done automatically by the EndNote style "Copernicus_Publications" for Word and will hopefully be fixed during type setting.

Line 557: The ERA5 reference is placed where it is not in right alphabetical order in the reference list.

This is also caused by the formatting of the EndNote style "Copernicus_Publications" omitting the authors of this website "Copernicus Climate Change Service (C3S)". This will hopefully also be fixed during type setting.

---

## Author Comment (AC2)

**Response to Reviewers**

**Geoscientific Model Development**

| | |
|---|---|
| Manuscript: | egusphere-2024-1518 |
| Title: | Monitoring and benchmarking Earth system model simulations with ESMValTool v2.12.0 |
| Authors: | Axel Lauer, Lisa Bock, Birgit Hassler, Patrick Jöckel, Lukas Ruhe, Manuel Schlund |
| Date: | 19 September 2024 |

*We thank the two reviewers for their helpful comments. In this document, we answer each point raised by the reviewers. The original reviewers' comments are given in **black**, our answers in blue. A "track changes" version of the revised manuscript highlighting all changes is available.*

**Reviewer #2**

This paper provides a description of some new functionality for ESMValTool. The ESMValTool is a key piece of software in the CMIP world, and these regular updates with new capabilities are extremely valuable. I am not a direct user of ESMValTool, so my comments here will be from a neutral perspective, but I will say that I'm more likely to become a user now that I've read this description. The paper is easy to read and understand, the topic is well within the scope of GMD by presenting new capabilities in a major software package and these new capabilities are substantial enough to warrant a publication. From my reading, there are a few things that could be made more clear. The main weakness in my opinion is the example problem that is used to show the new features seems both overly contrived and also not that compelling in terms of an example showing how to detect mistakes. I would not suggest getting rid of it entirely, but it might be useful to add another example. A few additional detailed comments are provided below.

We also thank Reviewer #2 for helping us to improve the manuscript. As we explain in more detail in our replies to the reviewer's comments below, we illustrate the new ESMValTool capabilities with an easy to perform and easy to understand test simulation that we conducted for this paper as an example. The approach of a single model simulation with an error introduced after five years seemed to us a more interesting showcase than a "normal" simulation (e.g. from an arbitrary CMIP6 model) as it allows us to illustrate the case of a problem occurring during run-time of a model and at the same time providing two datasets with and without a problem for comparison by splitting the simulation into two five-year periods. We think that very little would be gained from adding another simulation for the purpose of showcasing the new ESMValTool functionalities. In order to keep the paper short and the number of panels in each figure small, we would therefore prefer to not add a second simulation to the examples.

1. The text in the abstract (and elsewhere in the paper) makes some assumptions about how model development proceeds. Specifically, there is clearly and explicitly an assumption that 'historical' simulations are continuously produced during development. From my understanding, this is true for

some modeling centers, but not for all. Several centers I am aware of use the pre-industrial climate much more extensively during development. It is also important to note that some (maybe most?) model development happens in individual component models rather than in the coupled system. This is evident even in the EMAC example which uses an AMIP setup. The discussion in this paper is very strongly focused on diagnostics/metrics of the atmosphere; does ESMValTool work as well for other components (land, ocean, sea-ice, land-ice)?

We focus on the 'historical' simulations as for all metrics discussed in the paper a reference dataset is needed. Typically, such a reference dataset is based on observational data (observations or reanalyses). The vast majority of such observational data is available only for the historical time period, which is why we focus in this paper on such simulations. In principle, however, ESMValTool can use any dataset such as a well characterized pre-industrial control run as reference data. We make this clearer by now explicitly mentioning in the abstract that the extensions of ESMValTool described can also be used to compare e.g. a pre-industrial simulation to a suitable reference dataset.

While ESMValTool also works for components of the Earth system other than atmosphere, we deliberately chose some atmospheric examples to showcase the new ESMValTool functionality as the EMAC model used to produce the example simulation was operated in an atmosphere only (AGCM) setup. To make this clearer, we added the following sentence to the abstract:

"While the examples shown here focus on atmospheric variables, the new functionality can be applied to any other ESM component such as, for instance, land, ocean, sea-ice or land-ice."

2. Twice within the first several sentences (lines 30 and 39), biology/chemistry/biogeochemistry are mentioned. This early text heavily stresses the aspects of ESMs that go beyond GCMs/AOGCMs. The description raises the big issue of fitness for purpose of these models (line 36), and the need to evaluate them carefully. This is all good and true, but is somewhat disconnected from the rest of the paper which is focused on traditional physical quantities in the atmosphere. Without some example of metrics for chemistry/biology/biogeochemistry, I would suggest revising these first couple of paragraphs to more accurately set up what will be presented.

The metrics described in this paper can be applied to arbitrary variables from Earth System Model runs (please also see our answer to point 1). For example, instead of a time series of anomalies in near surface temperature (Figure 2), ESMValTool could be used to produce a time series of anomalies in Arctic September sea ice extent or ocean chlorophyll. Doing so would, however, in our opinion add very little to showcasing the new ESMValTool functionality as it would be the same metrics (bias, correlation, RMSE, EMD) simply applied to different variables. In order to keep the paper short, we would prefer to keep the number of reference datasets that need to be at least briefly introduced to a minimum. To make clearer that these are only examples and the new ESMValTool functionality can be applied to other ESM components, we modified the abstract (see our answer to point 1) and to section 3:

"We would like to note that the new ESMValTool functionalities shown in the following are not limited to atmospheric quantities but can be applied to any ESM component such as ocean, sea ice, land ice, etc. In case no suitable observationally-based reference dataset is available, also a well characterized reference model simulation can be used for assessing a simulation."

3. In Section 2.1, the ESMValTool is very briefly introduced. Too briefly, I think. While a detailed introduction to the software can be skipped, there are several things that would be useful if

described here. First are the system requirements for running ESMValTool. Second are the dependencies that ESMValTool needs. In terms of dependencies, maybe this could be an abbreviated discussion if there are many python packages needed, but it would be good to know if MPI, netCDF, etc. are required to be installed on the system, and if there are any compilers or languages that are needed.

ESMValTool does not have any specific system requirements except for a Unix style operating system that supports installation of cross-platform package managers such as Mamba (recommended) or Conda. As such, ESMValTool can even be run on a laptop (e.g. using the Windows Subsystem for Linux, WSL). The package manager is used to install all dependencies including, for instance, netCDF or programming languages such as R or Julia. There are no external compilers or system libraries needed.

Following the suggestion of the reviewer, we extended the description of ESMValTool in section 2.1 and also added some information on system requirements. As ESMValTool has about 700 dependencies, we only summarize the concept:

"ESMValTool can be run on any Unix-style operating system that supports installation of cross-platform package managers such as Mamba (recommended) or Conda. The package manager is used to install all dependencies including, for instance, netCDF or programming languages such as R or Julia. There are no external compilers or system libraries needed. Datasets available on the Earth System Grid Federation (ESGF) can be optionally downloaded automatically, for observationally-based datasets not available on ESGF, ESMValTool provides a collection of scripts with downloading and processing instructions to obtain such observational/reanalysis datasets."

4. At line 70, and again in the discussion, the use of wildcards is mentioned. Without having run ESMValTool, I was a little confused about this feature. It might be nice to have some kind of example to show what this really means for a user.

Following the suggestion of the reviewer, we added an example of how to use wildcards when running ESMValTool to section 2.1. The example shows how to use the first ensemble member (r1i1p1f1) from all available historical CMIP6 runs.

```
datasets:
  - project: CMIP6
    exp: historical
    dataset: '*'
    institute: '*'
    ensemble: 'r1i1p1f1'
    grid: '*'
```

5. There are just a couple of small things that I'd recommend for Section 2.2. Lines 90-91 state that the length of time intervals are used as weights. This sounds correct, but a little more clarification would be useful. If using monthly means, does this mean that the different lengths of months are used to correctly weight seasonal and annual averages? What metadata is needed for ESMValTool to determine the lengths (e.g., in cases where there could be leap years, or if a model uses a 360-day year)? Does the weighting deal with non-uniform time intervals outside of monthly means? For the spatial weighting, the text says it uses grid cell area, but it is unclear if that is calculated or provided by the user. This seems especially important for non-uniform grids where the grid cells may not be rectangular. The use of "absolute bias" is possibly a little confusing, especially later (around line 360) when the "absolute value of the absolute bias" is used. My solution, which might not be optimal,

would be to refer to X - R as the bias, and absolute bias as |X - R|. I wondered whether it would be easier to only present equations for the weighted metrics and remind the readers that normalized weights sum to unity and uniform weights sum to N?

The time weights are calculated from the input data using the bounds provided for each time step (variable "time_bnds") to obtain the length of the time interval. The individual time steps of the input data are then weighted using the lengths of the time intervals. As provision of time bounds is mandatory for a dataset to be compliant with the CMOR standard, this can be done for all input data. This method accounts for different calendars and years (e.g. leap year versus non-leap years).

For area weighting, the grid cell area sizes are used. In case of regular grids, area sizes can be either given as a supplementary variable specified in the ESMValTool recipe (typically CMOR variables "areacella" or "areacello") or calculated from the input data. In case of irregular grids, the grid cell areas must be provided as a supplementary variable. We added this clarification to Section 2.2.

Following the suggestion of the reviewer, we removed the equation for the unweighted metrics and we now refer to X - R simply as the bias, and absolute bias as |X - R|.

6. The description of the EMD mostly makes sense, but I wonder whether Equation 7 and the description that follows is actually the special case of the 1-dimensional Wasserstein metric? I was comparing with some resources on the web (e.g., wikipedia), and the description usually uses an infimum operator over the set of possible joint probably distributions (gamma). The description after the equation seems to have already done the minimization to get the optimal transport matrix. Maybe I'm just not understanding what the `min` operator in Equation 7 actually means? Maybe it is the minimization of all pdfs gamma in the nxm space? I'm not sure how the constraints are imposed, though, maybe only through the marginals? In which case, maybe this really is equivalent to the infimum. I did take a look at the Rubner et al. reference, but their description is in terms of the distance matrix and the flow matrix, and is similar to the other descriptions I looked at. Sorry, this is probably me being thick, but if Equation 7 could be made a little more clear in terms of the operator, that would be great. I guess, since I'm on about it, I might as well also ask how the distance is assigned in this implementation. Specifically, since the EMD is only applied to 1-d histograms, I wonder if the distance between every bin is just 1 or if it is the difference between each bin center value and every other bin center value?

The reviewer is right, what we describe here is a special case of the Wasserstein metric $W_1$ for 1-dimensional discrete distributions. Note that the subscript "1" does not refer to "one-dimensional", but rather to the exponent of the metric within the equation (very similar to p-norms).

The purpose of Eq. (7, now Eq. 5) was to provide an easy-to-understand and intuitive description of the EMD. From the various resources available online, we found the documentation of Pythons POT package (Python Optimal Transport; https://pythonot.github.io/quickstart.html#computing-wasserstein-distance) to be (arguably) the most intuitive one. They also use the minimum operator instead of the infimum operator in the definition of the Wasserstein metric. However, we could not find a scientific justification for this. For that reason, we decided to replace the minimum operator with the infimum operator in the revised version of Eq. (7, now Eq. 5), which is more general (each minimum is an infimum) and matches the definition used in Vissio et al. (2020) more closely. To make this clearer, we introduced the set of all joint distributions to the equation (instead of sums describing the marginals) and adapted the description accordingly. The paragraph now reads:

"With these probability mass functions, the EMD can be expressed as

$$EMD = \inf_{\gamma \in \Pi(p_x, p_r)} \sum_{i,j}^{n} \gamma_{ij} |x_i - r_j| \qquad (5)$$

Here, $\Pi(p_x, p_r)$ denotes the set of all joint probability distributions $\gamma$ with marginals $p_x$ and $p_r$. The sum describes the aforementioned transportation cost, which is proportional to the "amount of earth moved" (characterized by $\gamma$) and the "distance the earth has travelled" (characterized by absolute differences of the bin centers). The $\gamma$ that minimizes this transportation cost is called "optimal transport matrix"."

Regarding the actual implementation: for the simple 1-dim case we have, the EMD can be calculated analytically using the CDFs of $x$ and $r$ (see e.g. https://en.wikipedia.org/wiki/Wasserstein_metric#One-dimensional_distributions). However, since we think the revised Eq. (7, now Eq. 5) is more intuitive, we do not explicitly provide a formula for that, but just mention this as a side note in the manuscript:

"In practice, for our simple 1-dimensional case, the EMD can be calculated analytically with the cumulative distributions of $x$ and $r$ (see Remark (2.30) in Peyré and Cuturi (2019) for details)."

If we apply Eq. (7, now Eq. 5) directly, the distances would be the differences of the bin centers, but since these are equidistant, one could also simple assume a distance of 1 and scale the EMD appropriately.

7. A few questions about the datasets described in Section 2.3. First, I was going to ask if these datasets being distributed with ESMValTool, but then I saw at the end that they are not. It's nice that downloading and processing scripts are supplied; I imagine that some of those datasets require some kind of registration to get access? Are all the processing tools just using the same functions that are built into ESMValTool, or are they totally separate? The choices for these data are not really given, and all of them have several valid alternatives. Especially if these are being advocated as good reference data for use in evaluating ESMs, it might be good to add some words noting why these might be preferred over their alternatives. In the GPCP-SG data, I was not sure what the "SG" actually means. For the ISCCP-FH data, are there spurious trends through the late 1990s in the fluxes like in the ISCCP-H data?

The reviewer is correct, some of the datasets that can be used as a reference require, for instance, registration or acceptance of a license agreement before downloading and using the data. For this reason, datasets in ESMValTool are categorized into different "tiers" with tier 1 and tier 2 datasets freely available and tier 3 datasets with access restrictions. ESMValTool provides downloading and reformatting scripts for all datasets implemented that cannot be downloaded automatically from ESGF and used directly with the tool. For these scripts, all dependencies are installed when installing ESMValTool so that no other tools or libraries need to be provided.

For the examples shown in the paper, we selected reference datasets that are relatively commonly used to increase the possibility that the reader is already familiar with the dataset and only little description is needed. We do not advocate that these datasets are particularly suitable for specific applications or might be preferable over alternative datasets. We make this clearer in the revised version by adding the following sentence to Section 2.3.1:

"We would like to note that we do not advocate that these datasets used in the examples are particularly suitable for specific applications or might be preferable over alternative options."

"SG" in "GPCP-SG" is used to indicate that the dataset contains combined satellite-gauge (SG) data. The ISCCP-FH radiative flux profile data are based on the ISCCP-H cloud and atmosphere products. We are not aware of literature investigating possibly spurious trends in the ISCCP-FH data but it seems plausible that possible problems in ISCCP-H might translate into ISCCP-FH. Since no data from the late 1990s is used here and ISCCP-FH data are only used to provide an alternative dataset for CERES-EBAF in the portrait diagram shown in Figure 8, we do not expect this to be a problem here.

8. The EMAC experiment and the presentation of the example analysis did not seem all that convincing. It was surprising to me that replacing the SST with the zonal mean SST didn't lead to much, much larger errors. I wonder if some of the zonal asymmetries of the real world are being smoothed by the coarse resolution of EMAC (is the resolution stated?)? At line 290, the small error in seasonal cycle of SST is noted for the first 5-years in EMAC, but this isn't explained for readers who might not be familiar with these kinds of model experiments. The EMAC run is being forced with observed SST and ice, but is being compared to coupled runs that produce their own SST. So it isn't a fair comparison. Actually, I was wondering why Figure 3b shows such large RMSE given the prescribed SST? Is it because the reference data set is calculated over a different time interval, or is it because EMAC has some bias in 2m temperature that makes it more different from SST than in most models? The text seems to be following a line of thinking that his EMAC example is similar to what would be done during model development, but I don't think that a preliminary "quick look" analysis of a model run would split the dataset into time periods. I think a more plausible storyline for this analysis would be the user would take all 10 years of simulation and run ESMValTool on that, and only upon noting something that looks suspicious would they go back and start looking at different time periods. I'd suggest putting the metrics for the full 10 years on the plots.

It could well be that the rather coarse resolution of the example EMAC run (T42 resolution) smoothes some of the SST effects. We are aware that comparing an AMIP run to coupled (historical) runs is not an entirely fair comparison. This is mentioned explicitly in Section 2.1. We decided to use an AMIP type experiment as an example as this allowed for an easy and relatively robust way to introduce a model "error" during the run time of the simulation, which we found a more interesting example to showcase the new ESMValTool functionality than a "normal" model run. The idea was to be able to show that possible problems during a model simulation can be detected e.g. in the time series that can be used for monitoring simulations.

We are not sure what the reviewer means by "such large RMSE" in Figure 3b. The figure shows global averages including also all land surface points for which the model freely calculates the near-surface temperatures. In addition, EMAC did not use HadCRUT5 data to prescribe SST, which might also explain some deviations. Yet, the AMIP simulation clearly outperform the coupled simulations as one would expect.

We expect very little differences when showing two 10-year simulations (with and without modified SST) instead of two times 5-years of data, particularly since the model data are only used to showcase the new ESMValTool capabilities and not to analyze or evaluate this simulation. We decided to use only one simulation to keep the paper short while being able to demonstrate a model error occurring during run time in the time series plots (Figure 2). We clarified this by adding the following two paragraphs to the description of EMAC model and model simulation in Section 2.3.2:

"We would like to stress, that the examples in the following are not meant to assess the performance of EMAC but illustrate the new capabilities of ESMValTool with an easy to perform and easy to

understand test simulation only. Likewise, using historical simulations from CMIP6 models for comparison is an arbitrary choice and for the purpose of illustrating the examples only."

"The approach of a single model simulation with an error introduced after five years allows us illustrate the case of a problem occurring during run-time of a model (see section 3.1) and at the same time providing two datasets with and without a problem from the same model for comparison by splitting the simulation into two five-year periods (sections 3.2-3.6). Again, we would like to stress that this is meant as an example to illustrate the new capabilities of ESMValTool rather than analyzing or evaluating the EMAC simulation used."

9. An aspect of the example analysis that I thought was missing is a little more detail about the choices that the ESMValTool user needs to make. Are there decisions about how regions are defined (e.g., the tropics) and what land/sea mask to use, and temporal sampling (of test case versus reference)? How does that look in a user's "recipe"? Also, nothing is noted about how ESMValTool handles things when something doesn't work. What happens if the reference dataset doesn't match the specified time period? What happens if the CMIP model data download fails or is very slow? What happens if a specified variable is not available in the test case or the reference case? These are probably in the documentation, but it'd be nice to get an idea here to set some expectations.

The reviewer has a good point, but we feel that a detailed description of writing ESMValTool recipes and handling run-time errors would be beyond the scope of this paper as sufficient context on many technical aspects of ESMValTool would be needed. This would lengthen the paper considerably but not add much to the description of the new functionalities introduced here. Instead, we prefer to refer to the relevant literature, the extensive documentation and tutorial of ESMValTool describing the tool and all relevant technical aspects in detail. We therefore added references to the documentation and tutorial to section 2.1. In addition, a doi is provided for all example recipes shown in this paper:

"More information for users and developers of ESMValTool including how to write own recipes can be found in the documentation available at https://docs.esmvaltool.org. For new users, there is a tutorial available at https://tutorial.esmvaltool.org."

10. In the figures with stippling, are there options for how to handle the stippling? I'd imagine that most science applications would want to swap the stippling convention to emphasize where there is robust agreement among data sets. And there would also be users who would want to completely mask the stippled area instead of just obscuring it.

At the moment, there are no options to customize the stippling applied to map plots and zonal mean plots. At the moment, stippling is used to emphasize grid cells where there is agreement among the data sets, i.e. the calculated metrics is within a user-specified range from other models. Options to change this convention or the stippling style (including a simple masking) are good ideas that we are happy to include in the next update.

11. For monitoring a simulation, as discussed around line 367 and elsewhere and throughout the EMAC examples, does ESMValTool actually provide utilities for doing these time slices? That is, can the user specify to produce all diagnostics for particular time periods or time chunk sizes (e.g., 5 years)? Or is it the responsibility of the user to repeatedly apply ESMValTool to their data subsets?

The time periods to be analyzed are specified in the ESMValTool "recipes", small YAML scripts that specify input data, processing steps and diagnostics to be applied when running ESMValTool. For every dataset, it is possible to specify a start and an end time or time ranges (e.g. 5 years from the

first time step available). ESMValTool then extracts the requested time period before preprocessing the data and passing the results to the diagnostics. Typically, the whole simulation period is in one directory and ESMValTool looks through the files to find and extract the requested time period for a variable. The user does not need to process data with external scripts or provide other information than the start and end dates for the time slices.

12. For the portrait plot, are there options to specify how to display the results? It's easy to imagine use cases that would have more test cases than CMIP models. For example, one might want to put the CMIP results in one figure and all the test cases in another one? Can more than 2 reference data sets be used?

With the current implementation of the portrait diagram, up to two reference datasets can be used. Splitting the CMIP results and the test cases into different plots is probably not optimum, as the results are compared to the median error (or whichever metric is used). This means that it would be hard to compare the CMIP portrait plot with a test case only portrait plot as the underlying median error is different. In case there are many test cases, it would probably make sense to either simply create multiple portrait plots for subgroups of test cases or to exclude the results for all test cases from the calculation of the median error. This is possible via the preprocessor settings specified in the ESMValTool recipe.

13. During the wrap up of the paper, around line 448, the netCDF output from ESMValTool is mentioned. From previous reading, I think I remember that the ESMValTool produces plots and maybe some kind of browsable output? Maybe it would be worth mentioning what are the outputs of ESMValTool.

Following the suggestion of the reviewer, we added the following sentence to the description of ESMValTool in section 2.1:

"The output of ESMValTool typically consists of plots (e.g. png or pdf), netCDF file(s), provenance record(s), log files and an html file summarizing the output in a browsable way."